# Molecular and Cellular Characterization of Primary Endothelial Cells from a Familial Cavernomatosis Patient

**DOI:** 10.3390/ijms25073952

**Published:** 2024-04-02

**Authors:** Laura Lorente-Herraiz, Angel M. Cuesta, Jaime Granado, Lucía Recio-Poveda, Luisa-María Botella, Virginia Albiñana

**Affiliations:** 1Departamento de Biomedicina Molecular, Centro de Investigaciones Biológicas Margaritas Salas, 28040 Madrid, Spain; laura.lorente@cib.csic.es (L.L.-H.); jaime.granado@gmail.com (J.G.); lucia.recio@cib.csic.es (L.R.-P.); vir_albi_di@yahoo.es (V.A.); 2Centro de Investigación Biomédica en Red de Enfermedades Raras (CIBERER), ISCIII, 28029 Madrid, Spain; angcuest@ucm.es; 3Departamento de Bioquímica y Biología Molecular, Facultad de Farmacia, Universidad Complutense de Madrid, 28040 Madrid, Spain; 4Instituto de Investigación Sanitaria del Hospital Clínico San Carlos (IdISSC), 28040 Madrid, Spain

**Keywords:** cavernous malformations, cerebral cavernous malformation (CCM), cavernomatosis, vascular malformations, primary endothelial cells, splicing, CCM signaling complex, *Krit*-1

## Abstract

Cerebral cavernous malformation (CCM) or familial cavernomatosis is a rare, autosomal dominant, inherited disease characterized by the presence of vascular malformations consisting of blood vessels with an abnormal structure in the form of clusters. Based on the altered gene (*CCM1/Krit1*, *CCM2*, *CCM3*) and its origin (spontaneous or familial), different types of this disease can be found. In this work we have isolated and cultivated primary endothelial cells (ECs) from peripheral blood of a type 1 CCM patient. Differential functional and gene expression profiles of these cells were analyzed and compared to primary ECs from a healthy donor. The mutation of the familial index case consisted of a heterozygous point mutation in the position +1 splicing consensus between exons 15 and 16, causing failure in RNA processing and in the final protein. Furthermore, gene expression analysis by quantitative PCR revealed a decreased expression of genes involved in intercellular junction formation, angiogenesis, and vascular homeostasis. Cell biology analysis showed that CCM1 ECs were impaired in angiogenesis and cell migration. Taken together, the results obtained suggest that the alterations found in CCM1 ECs are already present in the heterozygous condition, suffering from vascular impairment and somewhat predisposed to vascular damage.

## 1. Introduction

Cerebral cavernous malformations (CCMs), also called cavernous angiomas or cerebral cavernomas, are vascular alterations consisting of clusters of abnormal blood vessels in the form of thin-walled caverns with no intervening stroma.

This type of malformation originates between veins and capillaries, with a lobulated appearance which, at a histological level, results in hypertrophied channels of endothelial cells (ECs). However, these lesions neither invade other tissues nor involve nervous cells. These abnormal structures have been described mainly in components of the central nervous system (CNS) such as brain, retina, and spinal cord and, sometimes, in other locations, such as ocular orbit, skin, and liver [1].

There are two forms of cavernomatosis: sporadic and familial. Sporadic cavernoma is not considered a rare disease since it affects 1 in 200–300 people [2]. It is not hereditary and appears in isolation. The familial form, familial cerebral cavernomatosis malformations (FCCM), which accounts for about 20% of total cavernoma cases, has a genetic basis and three genes are affected: *CCM1*, *CCM2*, and *CCM3*. FCCM shows an autosomal dominant inheritance with variable penetrance and an estimated prevalence of 1 in 5000–10,000 cases [2], making it a rare disease.

Clinical manifestations are variable. While some patients are asymptomatic, in other cases, cavernomas are detected by MRI (Magnetic Resonance Imaging) in medical screenings following seizures, migraines, epileptic episodes, or other neurological problems [3,4]. The most severe cases occur due to bleeding from cavernomatous lesions, which interfere with the functions of the CNS. Common symptoms are epileptic seizures and intracerebral hemorrhages, which in some cases result in brain infarctions and neurological deficits [5].

The molecular mechanisms causing the malformations are not yet completely elucidated. Cells from FCCM patients are heterozygous for the inherited mutation, but when a second mutation occurs in the normal allele, with loss of heterozygosity, the cavernoma lesions arise. Therefore, at the lesion site, mutations affect both alleles resulting in a lack, either physical or functional, of the protein encoded by the mutated gene.

FCCM 1 bears mutations in *CCM1/Krit* (7q21-22) and accounts for 53–65% of all cases; *CCM2/malcavernin* (7p13) accounts for 20% and *CCM3/PDCD10* (3q26.1) for 10–16% [6].

The proteins encoded by these genes (CCM1–3) form the CCM signaling complex (CSC), which interacts with different processes such as angiogenesis, endothelial cell–cell junctions (which seem to be the origin of the disease malformations), and different signaling pathways. Furthermore, the CSC also interacts with proteins involved in the regulation of endothelial structure such as Rap1 and HEG1 [7]. It also associates with proteins such as VE-cadherin and β-catenin, interfering with intercellular junctions [8].

The CSC actively participates in the RhoA-ROCK pathway. This pathway is involved in the anchoring between cells and with the extracellular matrix. It has been reported that CCMs are characterized by altered and unstable cell–cell junctions [9].

When the RhoA-ROCK pathway is active, ROCK phosphorylates the myosin light chain (MLC) and triggers a cascade of events, including the formation of stress fibers composed of actin filaments involving intercellular junctions. This process occurs in cell migration, angiogenesis, and cell proliferation [10]. Several studies support the idea that the CSC is directly involved in this process. Specifically, CCM1 associates with proteins that form cell–cell junctions such as β-catenin, which ultimately prevents the increase in ROCK activity. CCM2 inhibits this pathway as it favors RhoA degradation, and CCM3 recruits proteins involved in cell migration and adhesion, such as STK25 or Cdc42. In in vitro assays on altered CCM proteins from endothelial cells incubated with ROCK inhibitory drugs, the stress fibers decreased and the wild-type phenotype was recovered [9].

The present work constitutes a first study of molecular and functional characterization of ECs derived from endothelial peripheral blood precursors (EPCs) from a patient with familial cavernomatosis with *CCM1/Krit* harboring the mutation c.1563+1G>T; p.521sp (Figure 1).

The starting hypothesis is that familial cavernomatosis affects the expression pattern and the functionality of ECs, even in a heterozygous condition.

## 2. Results

### 2.1. Endothelial Cells from Primary Cultures of the CCM1 Patient and Mutation of the CCM1 Case

In the index case of this family a point mutation was found between exons 15 and 16 of the *CCM1/Krit1* gene: c.1563+1G>T; p.521sp, more specifically at the exon–intron consensus site at the end of exon 15, so the splicing process may be altered. This mutation has not been reported so far (Figure 1A, Sanger sequencing).

Figure 1B shows BOECs derived from the index case of the family and from a control donor. Although morphology is similar, in the CCM1 cell culture some cells are more elongated and show a different size (larger), with a more dispersed distribution in the culture plate. To test the presence of the mutation above in the BOECs of the patient, Sanger sequencing was carried out using primers around the intron splicing site as indicated in Section 4.

### 2.2. Analysis of Endothelial Gene Differential Expression between Control and Mutated Cells from a Patient with Familial Cavernomatosis Type 1

To analyze whether there is differential expression of endothelial genes between control and CCM1 cells, an analysis of RNA was performed by quantitative (real-time) PCR (qPCR). The genes chosen for this purpose were genes involved in the following endothelial functions: (i) angiogenesis: *Angiopoietin (ANGPT2*), *Endoglin (ENG)*, *VEGF-2 receptor* (*KDR*); (ii) cell-to-cell and cell-to-substrate adhesion: *PECAM1*, *CDH5*. ECM component: *SERPINE* (plasminogen activator inhibitor); (iii) endothelial physiology and vascular homeostasis: *NOS-3* and *PTGS2*. Cell cycle: *CCNB2*. Heterozygous mutated gene in the patient’s case: *CCM1*. Figure 2 shows the results of this analysis using *18S* RNA as a housekeeping gene. Results using *actin* as reference gene rendered the same pattern.

In the qPCR histograms, significant differences in expression between C-BOECs and CCM-BOECs are found in some genes. In CCM-BOECs, the expression of *CDH5* is significantly increased with respect to control, while *CCNB2*, *NOS-3*, *SERPINE-1*, and *PTGS2* are significantly decreased. Of interest, the levels of *CCM1* are similar in both samples. However, as the mutation affects splicing, we hypothesized that both mutated and wild-type transcripts were present, but the difference could be detected at the protein level. In fact, when a Western blot was carried out, the content of CCM1/KRIT protein dropped to around half in the cells of the patient compared to the control cells (Figure 3A). The signal of Krit protein was normalized using tubulin as loading control. Either β-actin or β-tubulin is commonly used as a loading control because their expression is relatively constitutive in most model systems. In our case, the CCM sample was slightly overloaded, as more tubulin signal was detected compared to control, however, the signal due to Krit was also weaker in the CCM1 sample.

### 2.3. CCM1 Endothelial Cells Proliferate Less Than Control Cells

The fact that *CCNB2* mRNA was downregulated in CCM1-BOECs (Figure 2) suggested an alteration in the proliferation rate compared to C-BOECs. Then, analysis of cell proliferation in CCM1 compared to control cells showed that, after 48 h of cultivation, the number of cells in CCM1 cultures was almost half of the number present in control cells (Figure 3B).

### 2.4. CCM1-BOECs Show Enhanced Actin Stress Fibers in the Cytoskeleton Compared to C-BOECs

It is known that CCM1 is part of the CSC involved in the RhoA-ROCK pathway in close connection with actin filaments constituting stress fibers. Thus, to assess whether the cytoskeleton of CCM1-BOECs was different from the wild-type (control) ECs, staining of actin and tubulin was performed and observed with fluorescent microscopy. In Figure 4 we show different images of actin stress fibers in CCM1 and control BOECs. The actin stress fibers are enhanced in the CCM1 heterozygous ECs compared to control cells. These results are in agreement with the role of the Rho family as an important regulator of the cellular cytoskeleton leading to impaired lumen formation and increased actin stress fibers [11].

On the other hand, the staining of tubulin was weaker in CCM1 than in control ECs (Figure 4). Gunel et al. [12] showed that CCM1/KRIT1 is a microtubule-associated protein with increased localization to microtubules, showing that KRIT1 and β-tubulin were present within the same complexes. Therefore, a loss of KRIT1 protein would lead to a lack of tubulin interaction and the decreased tubulin staining shown in Figure 4.

### 2.5. Functional Analysis in Control and CCM1-BOECs: Wound-Healing Tubulogenesis Assays

Having observed a different gene expression between control and CCM1 ECs, in particular genes important for endothelial functions, to assess the putative differential functionality of CCM1 and control BOECs, two assays were performed: tubulogenesis (an in vitro equivalent assay to measure angiogenesis) and “wound healing” (in vitro equivalent assay to measure the migratory capacity and wound repair).

In C-BOECs (Figure 5), after one and a half hours, cells begin to arrange themselves to form cell alignments with a geometric structure. Later, after 3 h, closed structures may be observed, with some remaining cells still in the tube formation phase. Finally, after 6 h, the predominant structure is a network of numerous completely closed tubules with thin walls consisting of a single row of cells.

In the CCM1-BOECs (Figure 5), the whole process seems to be delayed. After 1 h 30 min, the cells do not present such a differentiated formation as in the controls, with single/isolated cells still predominating. After 3 h, cells show an arrangement similar to control cells at an earlier stage (1 h 30 min), with few cells in completely closed structures. Finally, after 6 h, they present an arrangement similar to the picture taken after 3 h in control cells, not completely closed, and with the walls of these vessels formed by more than one cell. The number of completely closed structures was counted in five different fields at each time and conditions are represented.

The wound-healing assay aims to study cell migration. ECs tend to migrate to cover endothelial discontinuities that occur with wounds. Therefore, the wound-healing assay is an in vitro test to analyze the migration process by which ECs, in vitro, tend to cover the discontinuity generated in an endothelial monolayer and thus repair the wound.

In control cells at 3 h, the wound fronts had advanced by 21.8% compared to the initial measurement. Subsequently, at 6 h, the front had advanced by 50% compared to 0 h (Figure 6).

In the CCM1 cells, 3 h after wounding, the fronts have advanced by 17.6% compared to 0 h. Later, at 6 h, the fronts have not advanced, but it seems rather that they suffer a retraction with respect to 0 h, having advanced by 11.8% with respect to the beginning. The evolution in the discontinuity closure in control and CCM1 cells is shown within 6 h (Figure 6).

## 3. Discussion

CCM1 and its partners in the CSC are proteins involved in different signaling pathways/functions such as cell shape, cell proliferation, migration/wound healing, angiogenesis, and blood barrier permeability [13]. Therefore, the lack of CCM1/KRIT function leads to disruption of these signaling pathways and the already known clinical/pathological consequences.

The morphology and function of ECs are crucial as they are the first barrier between the blood and the different tissues. Despite biological and clinical studies, many questions remain unsolved about the dynamic biology of the endothelium and vasculature, as it is involved in a multitude of physiological processes in the whole body.

In our work, by culturing primary ECs (BOECs) from a healthy donor and from a patient with CCM1, we have tried to ascertain morphological, molecular, and functional differences in FCCM disease. These heterozygous cells from a CCM1 patient constitute the vessels of the body, excluding the cavernous lesions, where, after the second hit, no functional CCM1 protein is expressed. However, in a heterozygous condition we can already see differences with control cells. Thus, in the confluent state, control ECs hardly leave any empty intercellular space. They are held together by tight adherent junctions, which may be flexible to change in the process of modulating endothelial permeability and homeostasis. On the contrary, looking at the heterozygous CCM1 cells, it is apparent that some cells adopt a more irregular and elongated morphology, being stretched at one of the cell poles and oval at the other (Figure 1). These cells divide slower than control cells and do not reach complete confluence, leaving intercellular spaces among them. This observation suggests that the cell–cell junctions are somehow altered and that the partial lack of CCM1 protein is involved in this alteration.

The RNA expression analysis by quantitative PCR shows a significant increase in the expression of *CDH5*. This fact could be connected to the deficit in the CCM1 protein (Figure 2). If CCM1 is decreased, its interactions with proteins involved in cell–cell junctions such as β-catenin and VE-cadherin would be impaired. In such a situation [14,15], failure of appropriate positioning of VE-cadherin in the cell membrane results in EC permeability. Therefore, the increase in VE-cadherin transcription could be a way to compensate for the lack of CCM1, which through its interaction with β-catenin participates in intercellular junctions [16].

The decrease observed in the expression of the *PAI-1 (Serpine 1*) gene is explained by the increased signaling of KLF2/4. In fact, following the decrease in CCM1/KRIT1 [13], KLF2/4 downregulates *PAI-1 (Serpine 1*), as described in [17].

On the other hand, the alteration of intercellular junctions that we have seen in vitro would affect the permeability of the vasculature and its homeostasis in vivo. ECs act as a selective barrier that controls the movement of fluids, ions, and other macromolecules between blood and adjacent tissues by regulating junctional complexes. In addition, the endothelium regulates blood flow and tissue perfusion by modifying the diameter of blood vessels as well as vascular tone. ECs control vascular tone by producing NO, prostacyclin, and endothelium-derived hyperpolarizing factors, all of which are vasodilators. As evidenced by qPCR, NOS-3 expression is significantly decreased in CCM1 cells, which would result in reduced NO production and, ultimately, impaired modulation of vascular tone. Goitre et al. (2010) have shown that KRIT is part of the intracellular machinery that controls the redox balance [18]. In particular, these authors proved that the loss of KRIT1 led to a significant increase in the intracellular ROS levels. Importantly, KRIT1 regulates the expression of the superoxide anion (O_2_^−^) scavenging protein SOD_2_. Since O_2_^−^ can spontaneously react with nitric oxide (NO) to form OONO_2_, a decrease in KRIT may result in two deleterious effects, such as (i) a great reduction of NO bioavailability, with the consequence of impaired vascular tone and (ii) an increased formation of OONO_2_, a strong oxidant species with the potential to produce multiple cytotoxic effects, damaging the vasculature. As we have shown, NOS-3 expression is decreased in the endothelial cells of CCM1 patients, affecting the vascular tone and increasing the level of ROS species. Thus, normal vasculature in CCM1 patients may already be compromised to a certain extent. ECs have recently been reported to be compromised by local oxidative stress and inflammatory stimuli, which act as key pathogenic factors of CCM development [19,20].

In the tubulogenesis assays, endothelial CCM1 heterozygous cells take longer to align into tubes and to form closed structures compared to control cells. Since stabilization of intercellular junctions is required to form a closed tube, the delayed and defective angiogenesis may be a direct consequence of impaired cell junctions. The loss of at least half of the CCM1 protein prevents the CCM1 interaction with β-catenin/VE-cadherin. In fact, published molecular studies show that KRIT1 can regulate VE-cadherin-mediated endothelial cell–cell junction integrity [21].

The defective angiogenesis would lead to loss of capillaries, when levels of CCM1 are far below the theoretical 50% threshold corresponding to the heterozygous condition. Gunel et al. [12] showed that KRIT1 is a microtubule-associated protein. Immunoprecipitation experiments confirmed that KRIT1 and β-tubulin are present within the same complexes. One of the first stages of angiogenesis involves tube formation by endothelial cells. This tubulogenesis is triggered by interactions of endothelial cells with one another and with the extracellular matrix. Consequently, the link between KRIT1 and microtubules provides a potential pathway for signals from cell–cell contact and cell–matrix interaction, respectively, to influence cytoskeletal structure. Extrapolating from the heterozygous situation observed in the CCM1 ECs to a complete loss of CCM1 expression in cavernoma lesions, an abnormal capillary development resulting in impaired tubulogenesis would be expected, as shown in Figure 6. We believe that a weaker β-tubulin staining in endothelial cells from patients without a clear clinical presentation might be predictive of CCM1 diagnosis. Of course, more studies with different CCM1 mutations and families are necessary to support this hypothesis.

Whitehead et al. (2009) [22] provided the first in vivo evidence that any of the genes causing CCM were required in the endothelium. In CCM-depleted ECs, a loss of cortical actin with increased actin stress fibers was observed. In our experiments, ECs heterozygous for CCM1 show an increase in actin stress fibers. As discussed by Whitehead et al., these changes are typical of an activated RHOA GTPase pathway, which could be reversed by inhibitors of RHOA signaling. Cell culture data from Whitehead et al.’s study indicate that CCM2 regulates key aspects of the stabilized endothelium, including cellular architecture, barrier function, migration, and tube morphogenesis. Loss of CCM2 favors the destabilized phenotype. Normally, with two functioning alleles of CCM2, the intensity and duration of instability following an endothelial insult are limited. As seen in mice heterozygous for CCM2 [22] and likely in our heterozygous endothelial CCM1 cells, with only one functioning allele, there is a greater disruption of the stable state with increased permeability. The formation and maintenance of strong cell–cell contacts are favorable characteristics of ECs in a stable blood vessel. However, when injury, inflammation, or oxidative stress alters the endothelium, cell–cell junctions are temporarily disrupted and ECs initiate both migration and angiogenesis to repair the vessel. In this situation, CCM proteins will be required to recover the endothelium stability that will be mediated by limiting the RHOA activation [23].

The present work explores the possibilities of new avenues to personalize treatment for CCM patients. Personalized cultures from CCM patients would allow the study of not only the molecular characteristics of the culture derived from each patient but also the use of primary cultures for screening of drugs currently under trial for familial CCM [13] like fasudil or temol, inhibitors of the ROCK/Rho A pathway and of inflammatory processes, autophagy, and oxidative stress, respectively. Vitamin D has also been proposed as a therapeutical option [24]. Peripheral plasma vitamin D and non-HDL cholesterol reflect the severity of cerebral cavernous malformation disease. Atorvastatin (a statin) is also a ROCK inhibitor that contributes to normalization of the actin cytoskeleton [25,26]. Propranolol has also been used in vitro and in clinical trials to prevent bleeding, normalize blood pressure, and normalize vasculature, avoiding prothrombotic events [13,27,28].

In conclusion, the use of primary cultures of ECs from CCM patients would help to expand the idea of introducing personalized medicine in the study and characteristics of rare diseases, in particular when a treatment is necessary, which could be assayed in vitro using BOECs or/and cultures derived from surgery surplus of cavernous lesions.

## 4. Materials and Methods

### 4.1. Ethics

The entire procedure was approved by the ethical committee of the National Agency of Research in Spain (CSIC), with the reference 075/2017. Previous to the extraction, the patient (previously diagnosed with CCM1, mutation of the *CCM1/Krit1* gene: c.1563+1G>T; p.521sp) signed an informed consent form.

### 4.2. Human Samples: Blood Outgrowth Endothelial Cells’ Isolation and Cultivation

Blood outgrowth endothelial cells (BOECs) were grown from 50 mL peripheral blood, culturing buffy coat mononuclear cells on collagen-I-coated culture plates using EBM/EGM-2 culture medium (Promocell, Heidelberg, Germany), as previously described [29]. Briefly, cells from mononuclear layers were pelleted and resuspended in 5 mL of EBM/EGM-2 medium. Cells were centrifuged and the pellet washed twice in culture medium. Then, cells were resuspended in 5 mL of culture medium and plated on collagen-I-coated P-6 plates. Cells were incubated in 5% CO_2_ at 37 °C and humidity. Medium was replaced daily for the first week and, thereafter, every two days. BOECs were established as pure endothelial cultures, being the only surviving cell type covering the wells after 30–45 days of growth. The success of this technique ranges from 50% to 80%, depending on the patient, disease severity, type of collector tube, and anticoagulant used to collect the blood. Patients with active angiogenesis and blood collected in glass tubes with heparin or sodium citrate as an anticoagulant yield the highest success. The characterization is carried out either by flow cytometry or immunofluorescence with antibodies against specific endothelial markers such as CD31/PECAM-1 and von Willebrandt factor (vWF). This is carried out as a routine procedure. For characterization and functional studies, BOECs from the 2nd to 6th passage were used. From now on, we will refer to them as CCM1-BOECs.

At the same time, BOECs from a healthy donor were used as control for comparison, using the same range of passages as for CCM1-BOECs. We will name these C-BOECs.

Bright field and fluorescence microscopy images from the same samples were taken using a Zeiss Axiovert 135 microscope (Oberkochen, Germany). The FIJI-ImageJ 1.53q software tool (NIH, Bethesda, MD, USA) was used to process and quantify the fluorescence intensities.

### 4.3. RNA Extraction, Reverse Transcription, and Quantitative PCR

A pellet corresponding to around 3 × 10^5^ cells was subjected to RNA extraction using the NucleoSpin RNA purification kit (Macherey-Nagel GmbH&Co, Düren, Germany), following the manufacturer’s protocol. Around 1 μg of RNA was retrotranscribed using the Applied Biosystems kit (Thermo Fisher Scientific, Waltham, MA, USA).

Quantitative PCR was performed by FastStart Essential DNA Green Master (Roche, Basel, Switzerland) to amplify the following genes using the primers shown in Table 1. As housekeeping genes, 18S and β-actin were used. The ribosomal component 18S rRNA is a better internal control in RT-qPCR than β-actin and GAPDH because it shows less variance in expression across a variety of treatments and conditions. β-actin is also commonly used to normalize molecular expression studies due to its high conservation, but it has some limitations, as it may vary during growth, differentiation, and in disease states. Therefore, normalization with both is a double control. Samples were run in triplicate. Experiments were repeated at least three times.

### 4.4. Proliferation Assay

Proliferation of the control and CCM1-BOECs was measured following the “Luminescent Cell Viability Assay” (Promega, Madison, WI, USA) instructions. This is a homogeneous quantitative method to determine the number of viable cells in culture based on quantitation of ATP, which indicates metabolically active cells. Briefly, 5000 cells per well were seeded in quadruplicate in a collagen-coated P-96 plate. Cell Titer-Glo reagent (lysis buffer, Ultra-Glo Recombinant Luciferase, luciferine, and Mg^2+^) was added to wells to a final proportion of 1:1 and gently mixed for 30 min at room temperature (RT). Next, luminescence was measured in three independent measurements using a Glomax Multidetection System (www.promega.com/tbs/, 26 March 2024; Promega).

### 4.5. Western Blot

For protein extraction, BOECs, either C or CCM1, from 3 confluent P-6 wells were lysed on ice for 30 min in TNE buffer (50 mM Tris, 150 mM NaCl, 1 mM EDTA, and 0.5% Triton X100) supplemented with wide-range protease inhibitors (Roche) and lactacystin (Sigma-Aldrich, St. Louis, MO, USA), a specific proteasome inhibitor. Lysates were centrifuged at 14,000× *g* for 5 min. Similar amounts of protein from cleared cell lysates were boiled in SDS sample buffer and analyzed by 4–20% SDS-PAGE under reducing conditions (BioRad, Hercules, CA, USA). Proteins from gels were electro-transferred to nitrocellulose membranes (Amersham, Little Chalfont, UK) followed by immunodetection with a rabbit monoclonal anti-Krit1 (1:500 dilution) (Santa Cruz Biotechnology, Dallas, TX, USA) and antitubulin (1:5000 dilution) as loading control (Sigma-Aldrich). Following primary antibody incubation overnight at 4 °C, samples were washed and incubated with the corresponding horseradish-peroxidase-conjugated secondary antibodies from Dako (Glostrup, Denmark) at 1:2000 dilution, RT for 1 h. All antibodies were used at the dilution recommended by the manufacturer. Membranes were developed by chemiluminescence (SuperSignal West Pico Chemiluminescent Substrate, Thermo Scientific). Krit protein signal was quantified and normalized using tubulin in this case as a loading control. ImageJ software 1.53q was used for the densitometric quantification.

### 4.6. Immunofluorescent Microscopy

Immunofluorescence analyses were performed to visualize cytoskeleton in C- and CCM1-BOECs. To this purpose, a total of 50,000 cells were seeded on collagen-coated sterile coverslips (VWR international, Radnor, PA, USA) placed at the bottom of a 24-well plate. On the next day, cells were washed with PBS. Then, for staining actin filaments, cells were fixed, stained, and permeabilized in a single step by addition of 5 units/mL Alexa-546 phalloidin (Molecular Probes, Eugene, OR, USA), 100 μg/mL L-α-lysophosphatidylcholine, and 3.5% formaldehyde in cold PBS. Coverslips were mounted with Prolong+DAPI mounting media (Molecular Probes). Fluorescence images were taken using a confocal SP5 (DMI6000 CS Leica Microsystems, Wetzlar, Germany).

For tubulin/VE-cadherin staining, cells were fixed with 3.5% formaldehyde in PBS, washed, and blocked with 1% BSA in PBS for 1 h at 4 °C. Cells were incubated for 1 h at 4 °C with mouse antihuman tubulin/anti-VE-cadherin (1:100) (Sigma Aldrich). Following this, cells were washed thoroughly four times with PBS and incubated for 1 h at RT with goat antimouse IgG (H+L)–Alexa Fluor 568-conjugated antibody (1:200) (Thermo Fisher Scientific, Waltham, MA, USA). Finally, cells were washed with PBS and mounted and observed as described for actin.

### 4.7. Tubulogenesis: Endothelial Cell Tube Formation Assay

For this purpose, 50,000 cells per well, resuspended in 500 µL of EBM/EGM-2 medium, were seeded in a Matrigel-coated P-24 well (Thermo Fisher Scientific). Brightfield photographs were taken at different times between 0 and 6 h to count the number of completely closed cells.

### 4.8. Wound Healing

To study the migratory activity of ECs, we performed the wound-healing assay. For this purpose, 50,000 cells per well were seeded in a collagen-coated P-24 plate. Cells were incubated until confluence and, once a confluent monolayer was perceived, a “wound” (discontinuity) was made in the central area of the well. Then, brightfield images were taken at different times (0–6 h) to observe the advance of the front delimiting the initial discontinuity.

## Figures and Tables

**Figure 1 ijms-25-03952-f001:**
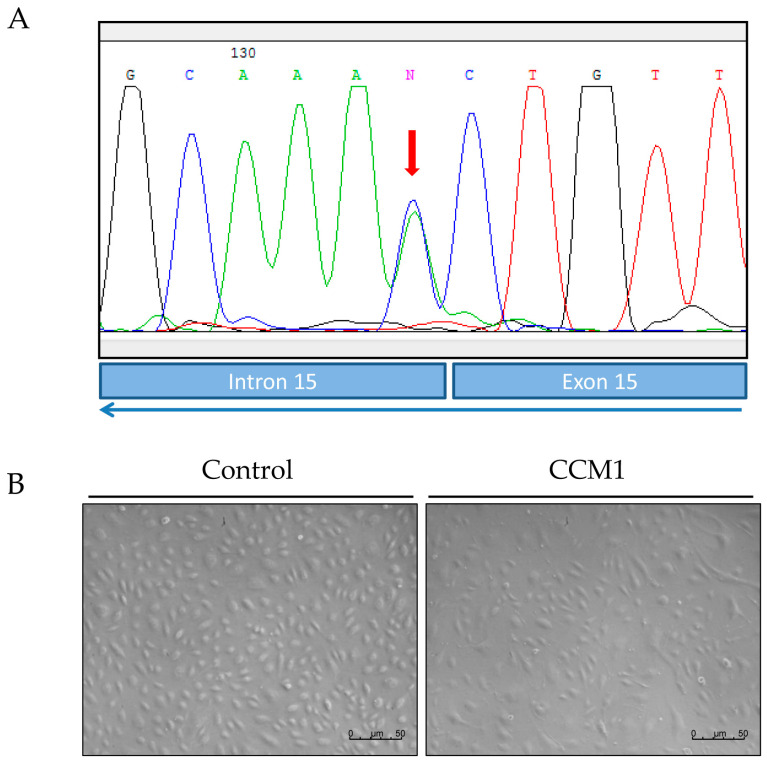
(**A**) Mutation of *CCM1/Krit1* gene; c.1563+1G>T; p.521sp, the reverse sequence is shown in the chromatogram. (**B**) Representative images of semi-confluent primary cultures of ECs derived from BOECs of a control donor (C-BOECs) (left side) and from the CCM1 patient (CCM-BOECs) (right side), whose cells carry the mutation shown in (**A**).

**Figure 2 ijms-25-03952-f002:**
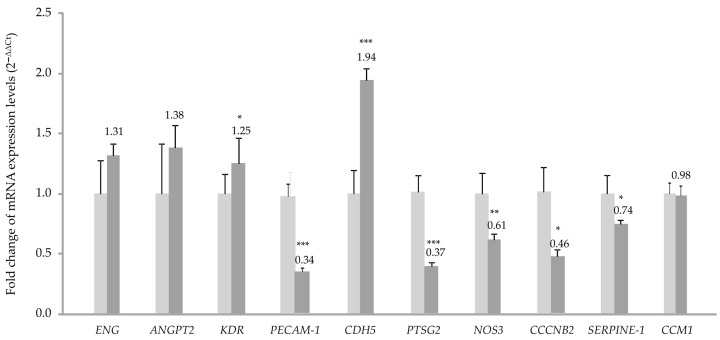
Real-time PCR. The differential expression of genes implicated in endothelial functions between C-BOECs (light grey) and CCM-BOECs (dark grey) is observed, using *18S* RNA as a housekeeping gene. Error bars denote ± SEM for each time. Student’s *t*-test: * *p* < 0.05; ** *p* < 0.01; and *** *p* < 0.001. The results represent triplicates in each PCR and qPCRs were repeated at least 3 times.

**Figure 3 ijms-25-03952-f003:**
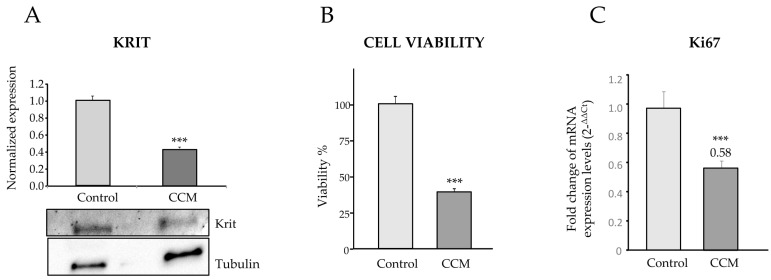
(**A**) Western blot analysis of Krit protein levels from BOECs, normalized to β-tubulin. Densitometric quantification ratios are shown. (**B**) Cell viability levels in C-BOECs (light grey) and CCM-BOECs (dark grey), showing a significant decrease in CCM samples. (**C**) Real-time PCR. Expression levels of Ki67. Error bars denote ± SEM. Student’s *t*-test: *** *p* < 0.001. The viability experiments were repeated 3 times and in each experiment quadruplicates were used.

**Figure 4 ijms-25-03952-f004:**
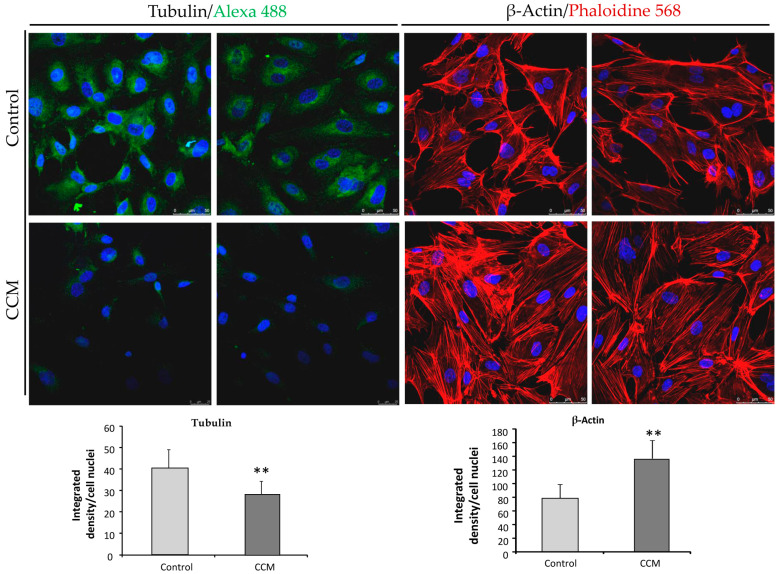
Representative confocal images using 63× objective of C-BOECs (light grey) and CCM-BOECs (dark grey) showing the cellular distribution of β-tubulin (Alexa 488, green) and actin (Alexa 568, red). Blue color corresponds to DAPI which marks nuclei. Quantification of total fluorescent signal normalized to the number of nuclei was performed in both cases and represented in the histograms shown below each type of staining. Error bars denote ± SEM. Student’s *t*-test: ** *p* < 0.01. The experiments were repeated 3 times and in each experiment duplicates were used.

**Figure 5 ijms-25-03952-f005:**
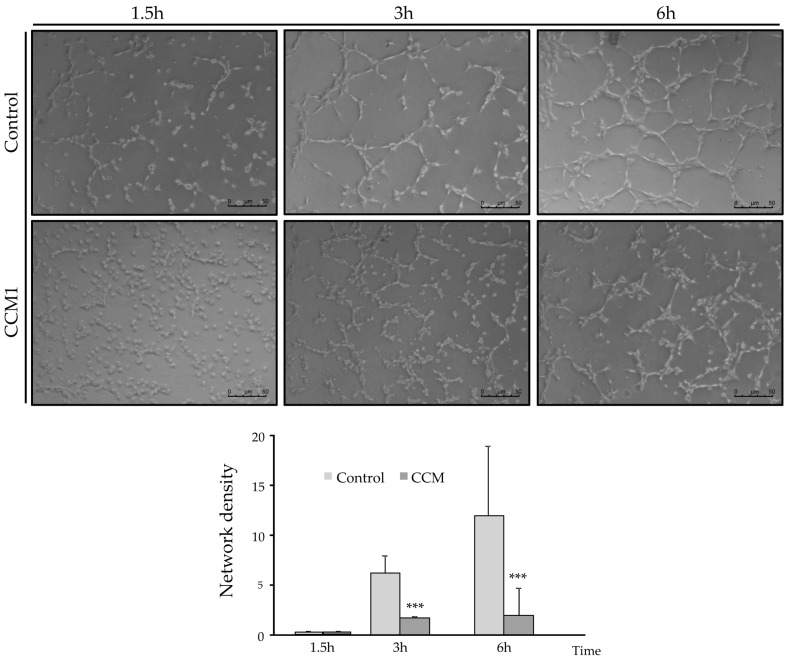
Endothelial Cell Tube Formation Assay of C-BOECs (light grey) and CCM-BOECs (dark grey). Representative images of the cells after 6 h incubation. It can be observed that the CCM BOECs show a lower number of totally closed tubes compared to the control cells. Student’s *t*-test: *** *p* < 0.001. Experiments were repeated at least three times with triplicates in each experiment.

**Figure 6 ijms-25-03952-f006:**
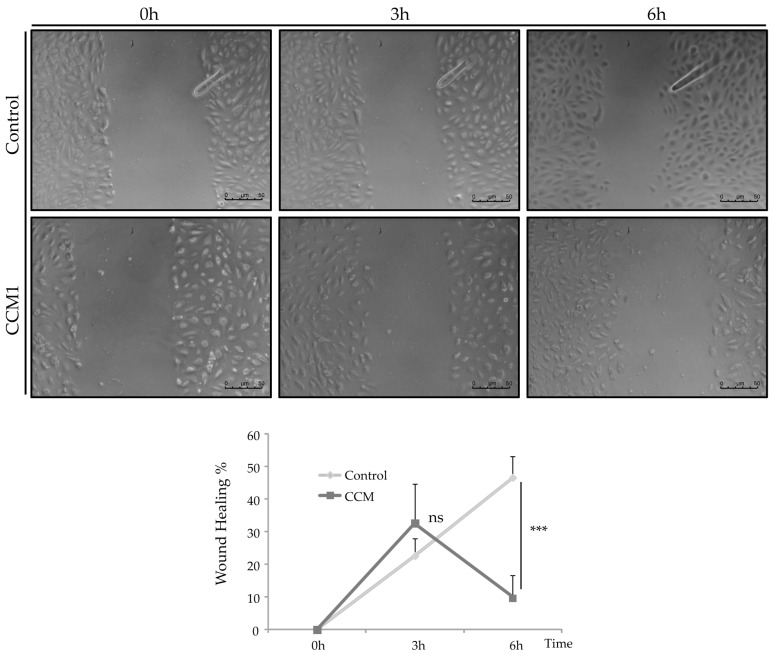
Wound-Healing Assay. Representative images of the cells after 3 and 6 h incubation. Analysis shows a significant difference in migration, indicating that wound repair slows down in CCM1-BOECs. Student’s *t*-test: ns stands for non-significant differences; *** *p* < 0.001. Experiments were repeated at least three times with triplicates in each experiment. C-BOECs (light grey) and CCM-BOECs (dark grey).

**Table 1 ijms-25-03952-t001:** Primers used for q-PCR assays.

Gene	Fwd 5′–3′	Rev 5′–3′
*18S*	CTCAACACGGGAAACCTCAC	CGCTCCACCAACTAAGAACG
*ENG*	AGCCACATCGCTCAGACAC	GCCAATACGACCAAATCC
*CCM1*	CTGTAAGAACATGCGCTGAAG	TCCATCGTACCTGTTACCAAAC
*ANGPT2*	TGCAAATGTTCACAAATGCTAA	AAGTTGGAAGGACCACATGC
*CCNB2*	TGGAAAAGTTGGCTCCAAAG	CTTCCTTCATGGAGACATCCTC
*PECAM-1*	AGAAAACCACTGCAGAGTACCAG	GGCCTCTTTCTTGTCCAGTGT
*PTGS2*	TCACGCATCAGTTTTTCAAGA	TCACCGTAAATATGATTTAAGTCCAC
*KDR*	GAGTGAGGAAGGAGGACGAAGG	CCGTAGGATGATGACAAGAAGTAGC
*NOS3*	GACCCTCACCGCTACAACAT	CCGGGTATCCAGGTCCAT
*CDH5*	GGAGGAGCTCACTGTGGATT	CTGATGCAGCAAGGACAGC
*SERPINE-1*	TCCAGCAGCTGAATTCCTG	GCTGGAGACATCTGCATCCT
*Ki67*	GAAAGAGTGGCAACCTGCCTTC	GCACCAAGTTTTACTACATCTGCC

## Data Availability

Data supporting the reported results can be found in our laboratory records. DNA and RNA are part of the collection associated with the research of the group.

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
