# Peer review of "Molecular and Cellular Characterization of Primary Endothelial Cells from a Familial Cavernomatosis Patient"

_ijms, 2024, doi:10.3390/ijms25073952_

Round 1

Reviewer 1 Report

Comments and Suggestions for Authors

Upon reviewing the manuscript titled "Molecular and Cellular Characterization of Primary Endothelial Cells from a Familial Cavernomatosis Patient," here are specific suggestions for improvement:

Results

  1. Gene Expression Analysis (Section 2.2):
    • The manuscript reports differential expression of genes related to angiogenesis, adhesion, and endothelial physiology between control and mutated cells. The significance of the expression changes in genes such as CDH5, CCNB2, NOS-3, SERPINE-1, and PTGS2 is noted. However, a more detailed interpretation of how these changes could affect endothelial cell function in the context of CCM1 pathology would enrich the discussion.
    • For Figure 2 (qPCR results), the clarity could be improved by adding specific fold-change values next to the statistical significance markers. This would provide a clearer immediate understanding of the magnitude of expression changes.
  2. Proliferation Assay (Section 2.3):
    • The decreased proliferation of CCM1-BOECs compared to control cells is a critical finding. It could be strengthened by including a comparative analysis of cell cycle phases using flow cytometry or another appropriate method to further elucidate the mechanism behind the reduced proliferation rate.
  3. Cytoskeletal Analysis (Section 2.4):
    • The observation of enhanced actin stress fibers in CCM1-BOECs and weakened tubulin staining could be complemented with quantitative analyses, such as the ratio of cells displaying these features or intensity measurements of fluorescence microscopy images. This would lend more precision to the claims.
    • Additionally, exploring the functional implications of these cytoskeletal changes on endothelial barrier function through permeability assays could offer deeper insights into CCM pathology.

Methods

  1. Isolation and Cultivation of BOECs:
    • While the manuscript describes the isolation and cultivation of BOECs from peripheral blood, including details like culture medium and plate coating, it could benefit from a discussion on the efficiency of endothelial cell isolation and purity assessment methods. Incorporating information on how endothelial cell identity was confirmed (e.g., through specific endothelial markers beyond the initial selection) would provide clarity.
  2. Gene Expression Analysis:
    • The manuscript mentions using 18S RNA and actin as housekeeping genes for normalization in qPCR experiments. It would be helpful to include a rationale for choosing these particular genes over others and to discuss any validation done to confirm their stability across the samples analyzed.
  3. Western Blot Analysis:
    • For the Western blot analysis, specifics on antibody dilutions and incubation times are provided, which is good practice. However, discussing the choice of β-tubulin as a loading control for Krit1 protein levels and whether any normalization or validation was done to ensure its consistent expression across samples would enhance the method's robustness section.

References

1.     Update the references to include the most recent studies in the field. In particular, consider including additional references to support the discussion and to provide context to the study’s findings. I suggest adding data related to recent bulk transcriptomics studies which could represent a strong substrate to enforce the role of described molecular mechanisms.

Comments on the Quality of English Language

The English should be revised.

Author Response

Upon reviewing the manuscript titled "Molecular and Cellular Characterization of Primary Endothelial Cells from a Familial Cavernomatosis Patient," here are specific suggestions for improvement:

Results

  1. Gene Expression Analysis (Section 2.2):
  • The manuscript reports differential expression of genes related to angiogenesis, adhesion, and endothelial physiology between control and mutated cells. The significance of the expression changes in genes such as CDH5, CCNB2, NOS-3, SERPINE-1, and PTGS2 is noted. However, a more detailed interpretation of how these changes could affect endothelial cell function in the context of CCM1 pathology would enrich the discussion.

Your comment is very good, and it helps to enrich the discussion and to put in value our qPCR results, we have added comments related to previous transcriptomic studies in the discussion shown in red

  • For Figure 2 (qPCR results), the clarity could be improved by adding specific fold-change values next to the statistical significance markers. This would provide a clearer immediate understanding of the magnitude of expression changes.

Thank you, we have added fold change with values on top of the bars in Figure 2

  1. Proliferation Assay (Section 2.3):
  • The decreased proliferation of CCM1-BOECs compared to control cells is a critical finding. It could be strengthened by including a comparative analysis of cell cycle phases using flow cytometry or another appropriate method to further elucidate the mechanism behind the reduced proliferation rate.

Thank you very much for the suggestion. We have no more cells at the moment of this culture, since it is a primary culture, not a cell line. However we will try to grow more as soon as possible. In the meantime, since we had RNA from the cells, we measured Ki67 by qPCR as a proliferation marker. The results have been included in the same figure as the proliferation assay (Fig 3C).

  1. Cytoskeletal Analysis (Section 2.4):
  • The observation of enhanced actin stress fibers in CCM1-BOECs and weakened tubulin staining could be complemented with quantitative analyses, such as the ratio of cells displaying these features or intensity measurements of fluorescence microscopy images. This would lend more precision to the claims.

Thank you for your suggestion, but we had this quantitative measures already included in Figure 4 (intensity measurements of fluorescence microscopy images, in the histograms lower part  of the figure 4).

  • Additionally, exploring the functional implications of these cytoskeletal changes on endothelial barrier function through permeability assays could offer deeper insights into CCM pathology.

Thank you very much for this suggestion, which is really interesting. We will follow your suggestion and are planning for a near future (function of endothelial barrier function) through permeability assays, when we will grow more primary cultures of CCM cells. At this moment, we need to grow more CCM cultures,

Methods

  1. Isolation and Cultivation of BOECs:
  • While the manuscript describes the isolation and cultivation of BOECs from peripheral blood, including details like culture medium and plate coating, it could benefit from a discussion on the efficiency of endothelial cell isolation and purity assessment methods. Incorporating information on how endothelial cell identity was confirmed (e.g., through specific endothelial markers beyond the initial selection) would provide clarity.

Thank you very much we have followed your advice and completed these details in the corresponding section:

 The success of this technique ranges from 50% to 80%, depending on the patient, severity of the   disease, and type of tube used to collect the blood. Patients with active angiogenesis, blood collected in glass tubes with heparin or sodium citrate as anticoagulants yield the highest success. The characterization is done either by flow cytometry or immunofluorescence with antibodies against specific endothelial markers such as CD31/PECAM-1 and vWF (von Willebrandt factor). This is done as a routine in the laboratory (data not shown).

  1. Gene Expression Analysis:
  • The manuscript mentions using 18S RNA and actin as housekeeping genes for normalization in qPCR experiments. It would be helpful to include a rationale for choosing these particular genes over others and to discuss any validation done to confirm their stability across the samples analyzed.

Thank you for the suggestion. We have included the following rationale in the manuscript:

The ribosomal component 18S rRNA is a good internal control in RT-qPCR because it shows less variance in expression across a variety of treatment and conditions. than β-actin and GAPDH. β-actin is commonly used to normalize molecular expression studies due to its high conservation, but it has some limitations, since it may vary during growth, differentiation, and in disease states. Therefore, normalization with both is a double control.

Western Blot Analysis:

  • For the Western blot analysis, specifics on antibody dilutions and incubation times are provided, which is good practice. However, discussing the choice of β-tubulin as a loading control for Krit1 protein levels and whether any normalization or validation was done to ensure its consistent expression across samples would enhance the method's robustness section.

Thank you for encouraging us to detail every thing. We have added in material and methods section the dilution used for each antibody, and also the way the quantification of Krit protein was made relative to beta-tubulin. In the results we have mentioned again the quantification normalized by beta tubulin as loading control. Beta-actin and beta-tubulin are commonly used as loading controls, because their expression is relatively constitutive in most model systems. In our case tubulin was used simply because the availability in the laboratory when the Western blot was done.

References

  1. Update the references to include the most recent studies in the field. In particular, consider including additional references to support the discussion and to provide context to the study’s findings. I suggest adding data related to recent bulk transcriptomics studies which could represent a strong substrate to enforce the role of described molecular mechanisms.

Thank you for your suggestion. We have included additional references, especially on recent transcriptomics to enrich the discussion. The additional discussion and references included are in red in the text.

Comments on the Quality of English Language

The English should be revised.

The English has been revised, thank you for raising the point.

Reviewer 2 Report

Comments and Suggestions for Authors

The authors performed gene expression analysis and established that CCM1/Krit1, CCM2, CCM3 may be involved in the different types of cerebral cavernous malformations and that CCM1 endothelial cells were already present in the heterozygous condition. The findings seem to be impressive and attractve for readers. Although the study scientifically sound well, I would like to male minor comment.

1. The authors investigted a role of tubulin abnormality in ECs among patients with cerebral cavernous malformations and proposed that alterations of CCM1 gene plays a pivotal role in the presentation of the disease. Please, add more information whether the overlape between other variants of tubulin abnormaloty might correspond to the disease and assist in prescreening these patients especially without clear clinical presentation of complete asymptomatic.

2. The mechanistic algorhytm for the procedure of the screening seems to be attractive for readers.

Author Response

Reviewer 2

The authors performed gene expression analysis and established that CCM1/Krit1, CCM2, CCM3 may be involved in the different types of cerebral cavernous malformations and that CCM1 endothelial cells were already present in the heterozygous condition. The findings seem to be impressive and attractve for readers. Although the study scientifically sound well, I would like to male minor comment.

  1. The authors investigated a role of tubulin abnormality in ECs among patients with cerebral cavernous malformations and proposed that alterations of CCM1 gene plays a pivotal role in the presentation of the disease. Please, add more information whether the overlape between other variants of tubulin abnormaloty might correspond to the disease and assist in prescreening these patients especially without clear clinical presentation of complete asymptomatic.

Thank you very much by your interesting comments. We believe that a loss of β-tubulin staining in endothelial cells from patients without a clear clinical presentation, may be associated to a CCM1 diagnostic. Of course more studies with different CCM1 mutations and families are necessary to support the assessment as of predictive value. In this sense, we have included the following paragraph in the discussion, marked in red, developing this idea.

The defective angiogenesis would lead to loss of capillaries, when levels of CCM1 are far below the theoretical 50% threshold corresponding to the heterozygous condition. Gunel et al [12] showed that KRIT1 is a microtubule-associated protein. Immunoprecipitation experiments confirmed that KRIT1 and β-tubulin are present within the same complexes. One of the first stages of angiogenesis involves tube formation by endothelial cell. This tubulogenesis is triggered by interactions of endothelial cells with one another and with the extracellular matrix. Consequently, the link between KRIT1 and microtubules provides a potential pathway for signals from cell–cell contact and cell–matrix interaction, respectively, to influence cytoskeletal structure. Extrapolating from the heterozygous situation observed in the CCM1 ECs, to a complete loss of CCM1 expression in cavernoma lessions, an abnormal capillary development resulting in impaired tubulogenesis would be expected, as shown in Figure 5. We believe that a weaker β-tubulin staining in endothelial cells from patients without a clear clinical presentation, might be predictive of CCM1 diagnostic. Of course more studies with different CCM1 mutations and families are necessary to support this hypothesis.

  1. The mechanistic algorhytm for the procedure of the screening seems to be attractive for readers.

Round 2

Reviewer 1 Report

Comments and Suggestions for Authors

The authors addressed all suggested points.

Comments on the Quality of English Language

The English is now ok.